# List Items One by One: A New Data Source and Learning Paradigm for Multimodal LLMs

**An Yan**$^\diamond$, **Zhengyuan Yang**$^\spadesuit$, **Junda Wu**$^\diamond$, **Wanrong Zhu**$^\heartsuit$, **Jianwei Yang**$^\spadesuit$, **Linjie Li**$^\spadesuit$,
**Kevin Lin**$^\spadesuit$, **Jianfeng Wang**$^\spadesuit$, **Julian McAuley**$^\diamond$, **Jianfeng Gao**$^\spadesuit$, **Lijuan Wang**$^\spadesuit$
$^\diamond$UC San Diego   $^\heartsuit$UC Santa Barbara   $^\spadesuit$Microsoft
{ayan,juw069,jmcauley}@ucsd.edu, wanrongzhu@ucsb.edu,
{zhengyang,jianwei.yang,keli,lindsey.li,jianfw,jfgao,lijuanw}@microsoft.com

## Abstract

Set-of-Mark (SoM) Prompting unleashes the visual grounding capability of GPT-4V, by enabling the model to associate visual objects with tags inserted on the image. These tags, marked with alphanumerics, can be indexed via text tokens for easy reference. Despite the extraordinary performance from GPT-4V, we observe that other Multimodal Large Language Models (MLLMs) struggle to understand these visual tags. To promote the learning of SoM prompting for open-source models, we propose a new learning paradigm: "list items one by one," which asks the model to enumerate and describe all visual tags placed on the image following the alphanumeric order of tags. By integrating our synthetic dataset with other visual instruction tuning datasets, we are able to equip existing MLLMs with the SoM prompting ability. Furthermore, we evaluate our finetuned SoM models on seven MLLM benchmarks. We find that this new dataset, even in a relatively small size (10k-30k images with tags), significantly enhances visual reasoning capabilities and reduces hallucinations for MLLMs. Perhaps surprisingly, these improvements persist even when the visual tags are omitted from input images during inference. This suggests the potential of "list items one by one" as a new paradigm for training MLLMs, which strengthens the object-text alignment through the use of visual tags in the training stage. Finally, we conduct analyses by probing trained models to understand the working mechanism of SoM. Our code and data are available at https://github.com/zzxslp/SoM-LLaVA.

## 1 Introduction

Recent advances in Multimodal Large Language Models (MLLMs) such as GPT-4V (OpenAI, 2023a) show strong performance in multimodal perception and reasoning, enabling various new capabilities (Yang et al., 2023b). Among these, Set-of-Mark Prompting (SoM) (Yang et al., 2023a) is an interesting new working mode that enhances the connection between visual objects and textual tokens via visual prompting, i.e., placing alphanumeric tags on input images. It provides a natural interface for human-computer interaction, by linking visual locations to executable actions through visual tags, and enables various applications such as GUI navigation (Yan et al., 2023b) and robot interaction (Lin et al., 2023a). Furthermore, GPT-4V with SoM (Yang et al., 2023a) can implicitly align visual objects with their corresponding tags. Such alignments (Li et al., 2020; Yang et al., 2021) allow MLLMs to leverage index numbers to perform multi-hop visual reasoning (Yang et al., 2023a; Wei et al., 2022), thereby improving their abilities in multimodal understanding and reasoning tasks.

Despite the significant interest in SoM prompting and its broad applications, it remains unclear why GPT-4V can benefit from SoM prompting, We find that other MLLMs, including the state-of-the-art open-sourced models such as LLaVA-v1.5 (Liu et al., 2024), and commercial systems like Gemini (Team et al., 2023), struggle to understand SoM prompts. This gap prevents them from leveraging the effectiveness of SoM prompting. In this study,

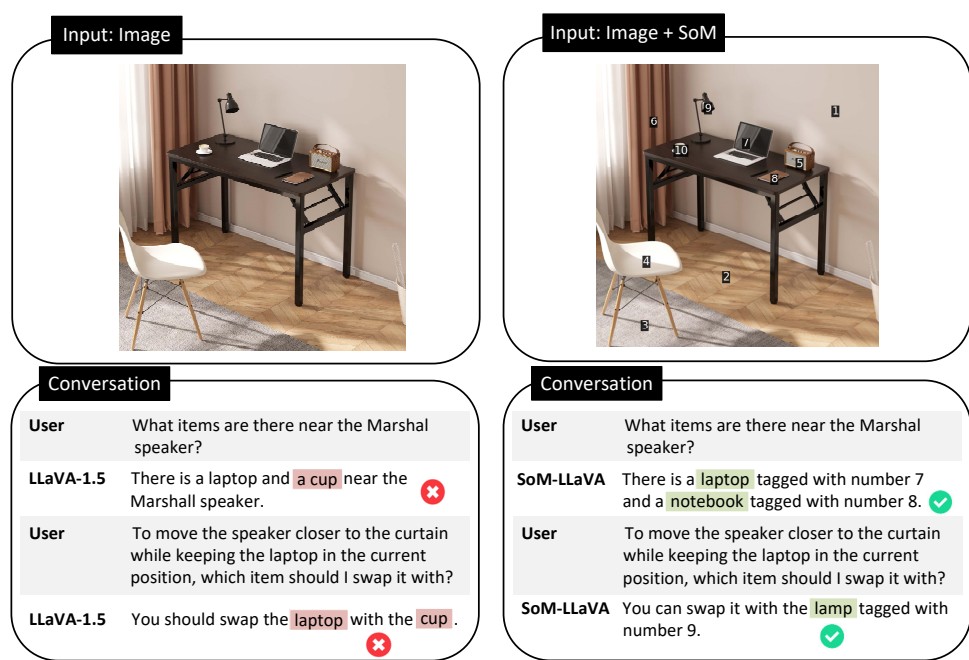

Figure 1: Example conversations from LLaVA and SoM-LLaVA (LLaVA with SoM ability) to demonstrate the effectiveness of our paradigm. **Left**: Standard prompting on LLaVA-1.5, which fails to correctly answer the questions. **Right**: Set-of-Mark prompting on SoM-LLaVA. Simply placing tags on the input image can improve visual reasoning of Multimodal LLMs.

we aim to deepen the understanding of SoM, with a goal of facilitating arbitrary MLLMs to benefit from it.

We break down SoM prompting into three core capabilities: (1) the ability to identify all tags and read the alphanumeric scene texts written on them; (2) the ability to recognize and pinpoint all objects in an image; (3) the ability to associate tags with corresponding objects in the image. Despite possessing skills such as OCR and visual recognition to meet the first two capabilities, most MLLMs still fail to fully understand SoM prompts. Therefore, we hypothesize that the crucial missing element is the third capability, associating tags with objects, which requires deliberate training. We further validate that SoM-style data are sparse in common MLLM training sources, and it may be necessary to create a specific dataset.

To facilitate such training, we introduce a new learning paradigm named "list items one by one". We show that by asking MLLMs to comprehensively list all tagged items following the alphanumeric order of visual tags, MLLMs can learn SoM prompting with a small number of item-listing samples. Specifically, we create a tailored synthetic dataset, by tagging images with Semantic-SAM (Li et al., 2023c; Yang et al., 2023a), and prompting GPT-4V to generate paired text descriptions. With just 10k image-text pairs, MLLMs like LLaVA-1.5 (Liu et al., 2023a) can reliably understand SoM tags. Based on this initial finding, we conduct studies to explore the effective recipes to help MLLMs best utilize SoM prompting.

We enhanced MLLMs with this "list items one by one" objective and assess their SoM performance from two aspects: model's ability to recognize and describe the SoM tags, and its ability to use SoM in improving multimodal reasoning ( Figure 1). For the first aspect, we design the tag listing task, which requires MLLMs to list and describe all tags in the image, evaluated by listing accuracy. For the second aspect, we evaluate finetuned models on seven MLLM benchmarks, including GQA, POPE, MME, SEED-Bench, LLaVA-Bench, MM-Vet and MMBench, showcasing that MLLMs with SoM can significantly boost the multmodal understanding performance. Moreover, our model trained with SoM data outperforms the original MLLM, even without additional visual tags during inference. This demonstrates the potential of incorporating our proposed dataset and learning paradigm to boost general MLLM training.

Finally, we revisit our original question regarding the working mechanism of SoM. The preliminary hypothesis is that the SoM capability may be related to OCR and the implicit association among text, tags, and objects. With our trained models, specifically SoM-LLaVA, we gain access to model features and attention maps for an in-depth analysis. We visualize the attention map to verify tag association. Compared with the original LLaVA model, SoM-LLaVA indeed learns better visual-tag-text associations, reflected in corresponding attention maps.

Our contributions are summarized as follows:

- We present a new training task and synthetic data source named "list items one by one", which effectively bootstraps MLLMs for the SoM visual prompting ability.
- We evaluate our finetuned models on general MLLM benchmarks, and show improved performance even when SoM tags are removed from the input image.
- We probe the working mechanism of SoM through the trained MLLMs, showcasing the implicit association between visual objects and text tokens when performing SoM prompting.

## 2 Related Work

**Visual referring prompting.** Other than text prompts, visual referring prompting (Yang et al., 2023b) is another effective approach when interacting with multimodal LLMs, where users directly draw on input images to specify their intent, such as drawing visual pointers or handwriting scene texts. Early studies show that vision-language models can understand visual pointers such as circles (Shtedritski et al., 2023) and dots (Mani et al., 2020). Recent studies (Yang et al., 2023b) show that more powerful multimodal LLMs (OpenAI, 2023a) can handle more complicated prompts such as arrows, boxes, circles, hand drawing, scene text, as well as their combinations. Another major advancement is Set-of-Mark Prompting (SoM) (Yang et al., 2023a), where numbered tags can be placed on images to associate visual objects with text indexed. Its effective visual grounding capability (Kazemzadeh et al., 2014; Yu et al., 2016; Mao et al., 2016) enables various applications (Yan et al., 2023b; Zhang et al., 2023). In this work, we aim to better understand SoM and extend its success from GPT-4V (OpenAI, 2023a) to other open-source multimodal LLMs.

**Multimodal LLMs.** Multimodal LLMs (Alayrac et al., 2022; Zhu et al., 2022; OpenAI, 2023a; Liu et al., 2023b; Li et al., 2023b; Xue et al., 2024) extend large language models (OpenAI, 2023b; Gao et al., 2023; Touvron et al., 2023) with visual perception capabilities. Recent studies (Chen et al., 2023) show the effectiveness of training open-source models on the GPT-4V generated detailed description data. Another thread of studies explore having multimodal LLMs predicting object locations as bounding boxes (Wang et al., 2023b; Peng et al., 2023) or masks (Rasheed et al., 2023). In contrast to most prior studies that pair the images with different text instructions, our study explores a new direction of how visual prompts such as SoM can improve multimodal LLMs. Specifically, we show that the SoM visual tags provide fine-grained alignments between visual objects and text tokens, thereby improving various visual reasoning tasks, both with and without SoM prompting during inference.

## 3 Preliminary Examination

### 3.1 Visualizing SoM Prompting on LLaVA

In this section, we first investigate the capacity of LLaVA-1.5 in SoM, concerning its attention sensibility to the numeric IDs tagged on the objects and its answer to the SoM query. We show an example task to list a series of objects tagged with numeric IDs in Figure 2, in which the attention map is extracted from LLaVA-1.5 based on the SoM query (*e.g.*, "I have labeled a bright numeric ID at the center for each visual object in the image. Please enumerate their names."). The top 20 image patches with the highest average attention weights across the user query tokens are highlighted in transparent red regions.

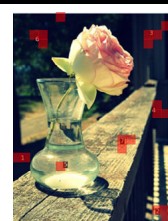

**LLaVA-1.5 Attention Map of Tags**

**User**: I have labeled a bright numeric ID at the center for each visual object in the image. Please enumerate their names.
**LLaVA-1.5**: In the image, there is a brown and black bird perched on a branch surrounded by leaves. The leaves surrounding the bird are also visible.

**LLaVA-1.5 Attention Map of Tags**

**User**: I have labeled a bright numeric ID at the center for each visual object in the image. Please enumerate their names.
**LLaVA-1.5**:
1. A vase with a flower in it.
2. A white flower in the vase.
3. The vase is on a wooden table.
4. The vase is sitting on a ledge.
5. The vase is sitting on a fence.
6. The vase is sitting on a railing.
7. … 8. … 9. … 10. … 11. …

Figure 2: Two examples of SoM prompting in LLaVA-1.5. **Left**: Attention map extracted from LLaVA-1.5 on the image of a bird perching on a branch, where 3 objects are tagged. **Right**: Attention map extracted from LLaVA-1.5 on the image of a vase placed on a table, where 7 objects are tagged. However, LLaVA-1.5 lists more than 7 object names that are repetitions of previous object names.

| # | Dataset | #Text | Text w/ Listing | Source of Text |
|---|---------|-------|-----------------|----------------|
| 1 | LLaVA-Pretrain-CC3M-595K | 595.4K | 0 | Raw CC3M image captions. |
| 2 | LLaVA-Pretrain-LCS-558K | 558.1K | 0 | Captioned by BLIP. |
| 3 | LLaVA-v1.5-Mix665K | 3356.2K | 0.72% | Rule-based, or generated by ShareGPT or GPT4-0314. |
| 4 | ShareGPT4V | 102.0K | 0.21% | Generated by GPT4-Vision. |
| 5 | CogVLM | 333.5K | 7.16% | Generated by MiniGPT4 or by GPT4-0314. |

Table 1: Examined pretraining (1–2) and instruction-tuning (3–5) datasets in our preliminary study.

We can observe from the highly attended regions of LLaVA-1.5 that the numeric ID tags can be easily and correctly attended by LLaVA-1.5 along with their associated objects (*e.g.,* bird, vase, and branches). Such capacities in locating numeric ID tags may have been acquired by LLaVA-1.5 from its pretraining tasks in OCR and also benefited from the strong OCR abilities of the ViT feature encoder (Radford et al., 2021) adopted by LLaVA-v1.5. However, the response prompted by the user query in the first example of Figure 2 suggests that LLaVA-1.5 cannot follow the SoM instruction to list all the items. Instead of providing the object descriptions corresponding to all the numeric ID tags, LLaVA-1.5 responds with a general image caption, due to a large portion of image captioning samples in its pretraining stage. From the second example of Figure 2, we can also observe that although LLaVA-1.5 generates a list of tag IDs with object names, it cannot accurately associate the tags to corresponding objects, causing the model to hallucinate the descriptions of these objects.

## 3.2 Finding SoM Data in Existing Training Sources

We further look into the pretraining/instruction-tuning (IT) dataset, aiming to inspect if there are text contents with listings, or images with SOM annotations. We examine the pretraining dataset of LLaVA-v1 and v1.5 (Liu et al., 2023b;a), and the IT dataset used by LLaVA-v1.5, ShareGPT4V (Chen et al., 2023), and CogVLM (Wang et al., 2023a).

Table 1 shows the source of text in each dataset and the percentage of text content with a listing format. The text in the two pretraining datasets for LLaVA are image captions (either the raw caption or generated by BLIP (Dai et al., 2023)), and we did not find any text with listings in them using our parser. Aside from image captions, the IT dataset also contains instructions related to other visual tasks such as VQA. We noticed that the answers provided by GPT-4(V) models sometimes construct the text in a listing manner (e.g., list out possible reasons for a question, list out observed objects in the image, etc). More examples can be found in Appendix A.6. The instruction-following dataset used by CogVLM has the highest percentage of text with listings (∼7%). Through our interaction with these models, we also find CogVLM is better at generating listing-style data than LLaVA-1.5.

We add tags to MSCOCO-2017 images following the SoM (Yang et al., 2023a) format, and train a binary classifier with ViT/B-16 (Dosovitskiy et al., 2020). We use the classifiers to filter the images in the two LLaVA pretraining datasets, and take the top 2k images with the highest scores for each dataset. We then manually check the top 2k images, and found 12 images with tagging in CC3M-595K (∼0.002%), and found 86 images with tagging in

LCS-558K ($\sim$0.015%). Figure 15 shows a few images with tagging. Given that tagged images are sparse in those datasets and the SoM prompting performance of open-source MLLMs is unsatisfying, it may be worthwhile to design a tailored dataset that empower open-source MLLMs with this emergent ability, similar to what GPT-4V is capable of.

# 4  Dataset Creation and Training

Motivated by the above analysis, in this section, we introduce the pipeline to create our dataset. First, in Section 4.1, we use semantic-SAM to generate semantic visual prompts in the form of numeric tags for each image. We then discuss the learning paradigm of "list items one by one" in Section 4.2. Finally, we use visual prompted images to generate text data in Section 4.3.

## 4.1  Image Source and Visual Prompting Generation

There are various open-source image datasets available  (Deng et al., 2009; Lin et al., 2014; Schuhmann et al., 2022; Yan et al., 2023a). We use MS-COCO (Lin et al., 2014) as the image source to create our SoM dataset, since it contains comprehensive human annotations with bounding boxes, masks, and captions. It has also been widely used for visual instruction tuning (Liu et al., 2023b; Wang et al., 2023a; Chen et al., 2023), which could benefit controlled experiments as well as comparisons with previous work.

The first step is to create visual prompts by placing numeric tags on proper locations. Following SoM (Yang et al., 2023a), we experiment with segmentation models including SEEM (Zou et al., 2023), Semantic-SAM (Li et al., 2023c), and SAM (Kirillov et al., 2023). Empirically, we find that Semantic-SAM provides the annotation granularity that best fits COCO images, and thus use it to create tagged images for our dataset.

## 4.2  A Learning Paradigm: List Items One by One

After obtaining the image data with semantic tags, the next question is how to design the instruction data to best distill the SoM visual prompting ability. A common approach (Liu et al., 2023b; Chen et al., 2023) in multimodal instruction-following data creation is to design and collect "question-answering" style samples. This is often done by prompting ChatGPT/GPT-4 or alternative open-source models. Given an image $I$ and optional metadata $M_I$ such as captions, bounding boxes, various questions or instructions $X_Q^{(i)}$ are posed, and the corresponding answers $X_A^{(i)}$ from large models are collected.

However, such general question-answering data may not be the most effective in distilling the desired SoM prompting capability, due to the inadequate mention of objects in text. For SoM prompting, one core ability of interest is to associate numbered tags with visual objects in the image, thereby enabling effective referral of visual objects via text tokens. In a general QA data, however, it is rare for multiple objects to be mentioned, even in an extended multi-turn conversation. To enhance tag association, we propose a simple and effective approach: *list items one by one*, where the model is asked to comprehensively describe all tagged items within an image. Given an image $I^\mathsf{T}$ with $N$ text tags on the image, we ask the model to enumerate all items in numerical order: $\{X_{obj}^1, X_{obj}^2, \cdots, X_{obj}^N\}$, where $X_{obj}^j$ is the textual description of the $j$-th item, tagged by ID $j$ in the image.

Beyond promoting SoM learning, *listing items one by one* is also effective in general multimodal LLM training: if a model learns to list items in the images with a specific order (in our case, the order is determined by the visual numeric tags), it gains a comprehensive and fine-grained understanding of images. This could directly benefit visual grounding and reasoning, which we verified through the standard multimodal QA and chat evaluation benchmarks.

Compared with existing visual instruction tuning datasets, such as LLaVA-665K (Liu et al., 2023a) and ShareGPT-4V (Chen et al., 2023), another difference is the implicit spatial infor-

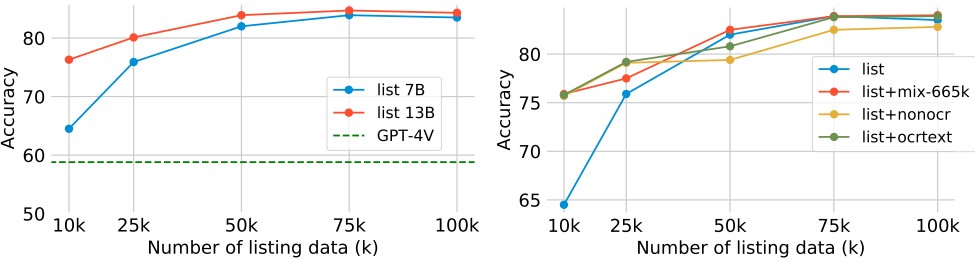

(a) Ablation on model sizes with LLaVA-1.5    (b) Ablation on data sources with LLaVA-1.5-7B

Figure 3: **Performance analysis on tag listing**. Training samples of listing data grow from 10k to 100k. *list+mix-665k* is to mix listing data with 665k instruction tuning data from (Liu et al., 2023a). *list+nonocr* is to exclude the OCR and text data from the full 665k data, resulting in 563k samples. *list+ocrtext* is to mix listing data with only OCR and text data from the full 665k data, resulting in 102k samples. Green-dashed line in Figure 3a is the zero-shot result from GPT-4V.

mation encoded by the visual tags in SoM prompting. Converting images into the language space inevitably loses information, especially spatial locations. For example, "a girl on the right" can only vaguely imply the position of the girl. However, with SoM visual prompting, we provide precise visual guidance on the image. Therefore, our data can be viewed as a form of dense captioning with a new way of encoding spatial information.

### 4.3 Text Data Generation via GPT-4V

With the visual prompting enhanced images, the final step for dataset creation is to generate the corresponding text data. To automate this process, we leverage GPT-4V (OpenAI, 2023a) to generate the listing data $\{X_{obj}^1, X_{obj}^2, \cdots, X_{obj}^N\}$, following the order of visual tags in the images. However, we find that simply prompting the model to list items in a zero-shot manner could lead to noisy and biased generation results, where the model may refer the tag to a distant object that is easy to describe. (see examples in appendix A.4). To mitigate this problem, we seek two complementary solutions: (1) We modify the system message of GPT-4V to avoid assigning tags to distant objects. (2) We manually design a few correct listing samples via human annotations, and use them as seed examples for in-context-learning to query GPT-4V. The details of our template is in Appendix.

In addition to listing, we also consider conversational data similar to LLaVA (Liu et al., 2023b), where GPT-4V is asked to generate multi-turn conversations between an AI assistant and a person asking questions about the photo. Given a tagged image $I^T$, we use GPT-4V to generate instruction-following data in the form of $\{$Person:$I^\mathsf{T} X_Q^{(i)}$, Assistant: $X_A^{(i)}\}$.

### 4.4 Model Training

We take the pretrained stage of LLaVA-1.5 (Liu et al., 2023a) as the base model, and continue finetuning by mixing instruction tuning data of LLaVA-1.5 with our collected visual prompting data. For SoM-listing, we create 40 task templates as human instructions (e.g., "please enumerate object names in the tagged image"), and treat them as standard conversational data. We use the same training objective of next-token prediction to train general QA, SoM-QA and SoM-listing data. Specifically, we maximize the conditional log likelihood as follows:

$$-\log p(X_A|X_v, X_Q) = -\log \prod_{i=1}^{L} p_\Theta(x_i|I/I^\mathsf{T}, X_{Q,<i}, X_{A,<i}), \tag{1}$$

where $\Theta$ are the trainable model parameters, $X_{Q,<i}$ and $X_{A,<i}$ are the instruction and answer tokens in all previous turns of conversations before the current prediction token $x_i$. The input image is $I$ or $I^\mathsf{T}$ for LLaVA or SoM data, respectively.

# 5 Experiments

## 5.1 Experimental Settings

**Experiment overview.** We validate the method effectiveness from two aspects. First, in Section 5.2, we benchmark the model's capabilities in understand and describing SoM visual prompting. We design the tag listing task on MS-COCO to test the SoM performance. Second, in Section 5.3 and 5.4, we evaluate if our dataset and model can benefit visual reasoning tasks, where we consider seven representative visual question answering and reasoning tasks detailed as follows.

**MLLM benchmarks.** We consider the following multimodal LLM benchmarks in Table 2 to validate SoM visual prompting's benefit on general visual reasoning tasks. GQA (Hudson & Manning, 2019) focused on fine-grained compositional reasoning over real-world images. We compute match accuracy following (Liu et al., 2023a). POPE (Li et al., 2023e) is to evaluate object hallucination in multimodal LLMs. We follow POPE and report the F1 Score for the binary choice questions. MME (Fu et al., 2023) contains 2800 binary choice questions for perception and cognition evaluation. We report the overall perception score for the evaluated models. SEED-I (Li et al., 2023a) contains 14K multiple choice questions on images. MMBench Liu et al. (2023c) is another multi-choice benchmark to evaluate the multi-modal understanding capability of MLLMs. We report the multiple choice accuracy on SEED-I and MMBench dev set. LLaVA-W (LLaVA-Bench In-the-Wild) (Liu et al., 2023b) and MM-Vet (Yu et al., 2023) are open-ended generation tasks, which compute the evaluation score by prompting a GPT-4 based evaluator (OpenAI, 2023b) with both the predicted and ground-truth reference answer. The score is then scaled to the range of 0 to 100.

We provide more implementation details in appendix A.1.

## 5.2 Evaluation on Tag Listing

First, we evaluate model performance on the tag listing task, aiming to answer two research questions: (1) Do model sizes matter in terms of learning SoM ability? (2) How will different sets of extra training data impact the SoM performance? We design the listing data based on images with ground-truth mask annotations from MS-COCO, and enumerate each object with corresponding class name. An example list is "1. person, 2. cat, 3. dog.". We compute list-wise accuracy, where for a caption with $N$ items, the score is $\frac{M}{N}$ with $M$ items predicted correctly by the model. With human annotation of objects in an image, we can automatically create abundant rule-based data (up to 100k) for studying model behaviors and perform quantitative evaluations.

For the first question, we find that larger LLM performs better for the listing task (see Figure 3a), presumably benefiting from the stronger language prior to help learn SoM prompting. For the second question, we decompose the 665k instruction data from LLaVA-1.5 (Liu et al., 2023a) into two parts. We find that both general caption-QA data, as well as OCR-text data contribute to learning SoM ability when limited listing data are available (10k). The reason could be that OCR can help with identifying numeric tags, and general caption may help the model to recognize objects within an image, both of them are fundamental abilities required by SoM. In general, other visual instruction data may benefit learning SoM, especially when SoM data is scarce.

Overall, we observe that with only 10k data, we can outperform zero-shot GPT-4V in listing accuracy, whereas growing data size from 50k to 100k only slightly improves the listing performance. These findings suggest that collecting a small amount of data may be sufficient for learning SoM prompting.

## 5.3 Evaluation on MLLM Benchmarks

We then train LLaVA-1.5 on our collected dataset and perform evaluation on MLLM benchmarks. As shown in Table 2, we observe that our SoM-LLaVA-1.5, which is trained with a mixture of LLaVA visual instructions and our SoM data in order to learn SoM prompting,

| Method | LLM | Res. | Pre-Data | IT-Data | POPE | MME | SEED-I | LLaVA-W | MM-Vet |
|--------|-----|------|----------|---------|------|-----|--------|---------|--------|
| BLIP-2 | Vicuna-13B | 224 | 129M | - | 85.3 | 1293.8 | 49.7 | 38.1 | 22.4 |
| InstructBLIP | Vicuna-7B | 224 | 129M | 1.2M | – | – | 58.8 | 60.9 | 26.2 |
| InstructBLIP | Vicuna-13B | 224 | 129M | 1.2M | 78.9 | 1212.8 | – | 58.2 | 25.6 |
| Fuyu-8B | Fuyu-8B | 600 | – | – | 74.1 | 728.6 | – | – | 21.4 |
| LLaMA-Adapter-V2 | LLaMA2-7B | 336 | – | – | – | 1328.4 | 35.2 | – | – |
| mPLUG-Owl-2 | LLaMA2-7B | 448 | 348M | – | – | 1450.2 | 64.1 | – | _36.2_ |
| Qwen-VL | Qwen-7B | 448 | 1.4B[†] | 50M[†] | – | – | 62.3 | – | – |
| Qwen-VL-Chat | Qwen-7B | 448 | 1.4B[†] | 50M[†] | – | 1487.5 | 65.4 | – | – |
| SPHINX | LLaMA2-7B | 224 | - | - | 80.7 | 1476.1 | 69.1 | _73.5_ | 36.0 |
| **LLaVA-1.5** | Vicuna-13B | 336 | 558K | 665K | 85.9 | 1531.3 | 68.2 | 70.7 | 35.4 |
| **SoM-LLaVA-1.5** | Vicuna-13B | 336 | 558K | 695K | _86.6_ | _1563.1_ | **69.6** | **75.3** | 35.9 |
| **SoM-LLaVA-1.5-T** | Vicuna-13B | 336 | 558K | 695K | **87.0** | **1572.8** | _69.5_ | 73.3 | **37.2** |

Table 2: **Performance comparison on popular MLLM benchmarks.** Res., Pre-Data, IT-Data indicate input image resolution, the number of samples in pretraining and instruction tuning stage, respectively. [†]Includes in-house data that is not publicly accessible. Underlined numbers are the second best results in the column. SoM-LLaVA-1.5-T is the model with tagged images as input.

| Method | LLM Size | GQA | POPE | MME | SEED-I | LLaVA-W | MM-Vet | MMBench |
|--------|----------|-----|------|-----|--------|---------|--------|---------|
| **LLaVA-1.5** | 7B | 62.0 | 85.9 | 1464.0 | 64.8 | 63.4 | 30.5 | 65.4 |
| **SoM-LLaVA-1.5** | 7B | **62.7** | **86.5** | **1507.0** | **67.0** | **66.9** | **33.3** | **66.5** |
| **LLaVA-1.5** | 13B | 63.3 | 85.9 | 1531.3 | 68.2 | 70.7 | 35.4 | 68.9 |
| **SoM-LLaVA-1.5** | 13B | **63.8** | **86.6** | **1563.1** | **69.6** | **75.3** | **35.9** | **69.5** |

Table 3: **Performance comparison on LLaVA models.** We reproduced results of LLaVA-1.5 on MME and MMBench-dev with the official checkpoints and evaluation scripts.

also obtains superior performance on general MLLM tasks. Surprisingly, we find that even without tagged images, SoM-LLaVA still attains strong performance and substantial improvement over the orignal LLaVA. This indicates the quality of our data and the potential of introducing listing data into general MLLM training to improve visual understanding and reasoning, as well as reduce hallucinations. We conjecture the reason that the great performance of SoM-LLaVA on non-tagged images is that "listing items one by one" with visual prompting guides the model to learn fine-grained semantics for image features. Related case studies and visualizations are in appendix A.2. For the performance of open-vocabulary listing, we present examples in appendix A.3.

## 5.4 Additional Evaluation Results on LLaVA

We evaluate LLaVA and SoM-LLaVA models with 7B and 13B LLM backbones, to further demonstrate the effectiveness of adding SoM data into the instruction tuning stage. As shown in Table 3, we observed performance improvements across a wide range of benchmarks testing multimodal understanding and reasoning capabilities. These gains are consistent on both model architectures.

## 5.5 Ablation Study on Mixture of Datasets

Finally, we perform ablation on different data mixture strategies in Table 4. We consider mixing our listing and QA data generated from Section 4.3 with LLaVA-665k (Liu et al., 2023a), trained separately or together. Empirically, we find that mixing listing and QA data yields the best overall performance. In Section 5.2, we find OCR data can help the learning of listing. Here we also notice that "listing item one by one" can in turn greatly improve the performance of OCR related task. The results on POPE indicates our data leads to lower hallucinations compared with ShareGPT-4V, which is a dense caption dataset without visual prompting. Placing tags on the images can seamlessly encode spatial information into the data for MLLMs to learn fine-grained vision language alignment.

| Data Composition | Data Size | POPE | | | MME | | SEED-I |
| --- | --- | --- | --- | --- | --- | --- | --- |
| | | random | popular | adversarial | OCR | overall | overall |
| LLaVA-IT | 665K | 87.1 | 86.2 | 84.5 | 125.0 | 1531.3 | 68.2 |
| LLaVA-IT + Listing | 665K + **10k** | 87.3 | 86.3 | 84.8 | **147.5** | **1588.2** | 68.9 |
| LLaVA-IT + QA | 695K + **20k** | 87.5 | 86.4 | 84.7 | 110.0 | 1540.0 | 69.2 |
| LLaVA-IT + Listing + QA | 695K + **30k** | **87.8** | **86.7** | **85.2** | 140.0 | 1563.1 | **69.6** |
| LLaVA-IT + ShareGPT-4V | 695K + **20k** | 87.1 | 86.0 | 84.3 | 110.0 | 1528.7 | 69.3 |

Table 4: **Comparison for different data mixture strategies.** LLaVA-IT is the mix665k data from (Liu et al., 2023a). Listing and QA is from our SoM dataset with tagged image-text pairs. ShareGPT-4V is from (Chen et al., 2023) with the same MS-COCO images as our 2k QA data and detailed captions from GPT-4V. Results are evaluated on the 13B LLM.

# 6  Analysis

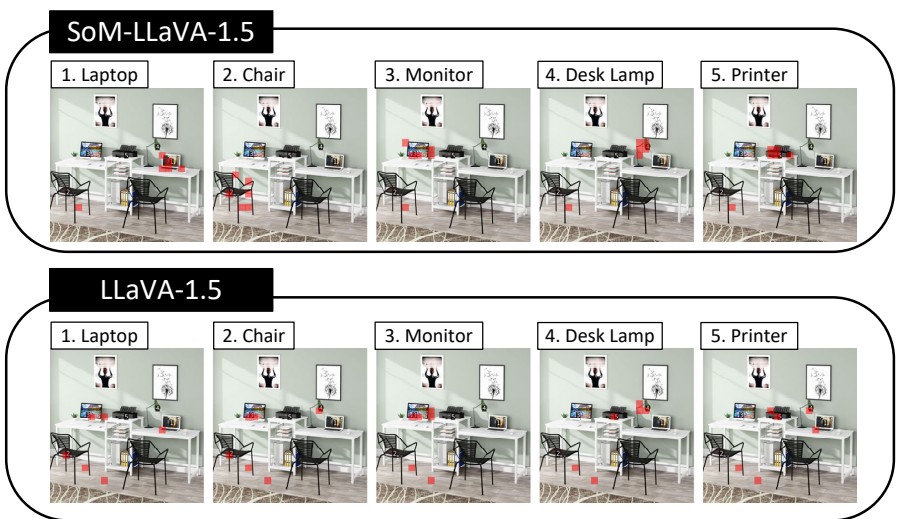

Figure 4: **A comparative example of attention maps extracted from LLaVA-1.5 and SoM-LLaVA-1.5**, where five objects (*e.g.*, laptop, chair, monitor, desk lamp, and printer) are tagged. We highlight the top-5 most attended image patches of the models on each object's numeric tags individually. SoM-LLaVA is better at attending to objects following numeric text and tags.

## 6.1  Probing Trained Models

We first analyze the tag-listing capacity of SoM-LLaVA-1.5 (13B) acquired through fine-tuning. In Figure 4, we show the attention maps on the five tagged objects, which are extracted from SoM-LLaVA-1.5 and LLaVA-1.5 respectively. The comparative example showcases that although both models can locate their model attention on the mentioned objects to some extent, the fine-tuned SoM-LLaVA-1.5 model can attend to and focus on characteristic regions of the object, which can also be accurately guided by the numeric ID tags. For example, the comparative attention maps on the object "Laptop" tagged with number 1 show that SoM-LLaVA-1.5 can clearly attend to the mentioned object with its main focus. In contrast, LLaVA-1.5 mistakenly attends to the monitor instead of the laptop, due to high similarity between these two objects.

In addition, we also observe that SoM-LLaVA-1.5 can be efficiently guided by the numeric ID tags to focus on the specific object the user refers to, even with multiple similar objects within the image. For example, the attention map of SoM-LLaVA-1.5 on the "Chair" tagged with a number 2 is mostly focusing on the chair on the left-hand side, instead of the similar chair on the right-hand side. SoM prompting in SoM-LLaVA-1.5 with such the capacity to accurately locate the tagged object, enables more flexible and easier user-referring queries without complicated language descriptions. The attention maps also verify our early hypothesis regarding the implicit association among the text, tag, and object in SoM prompting.

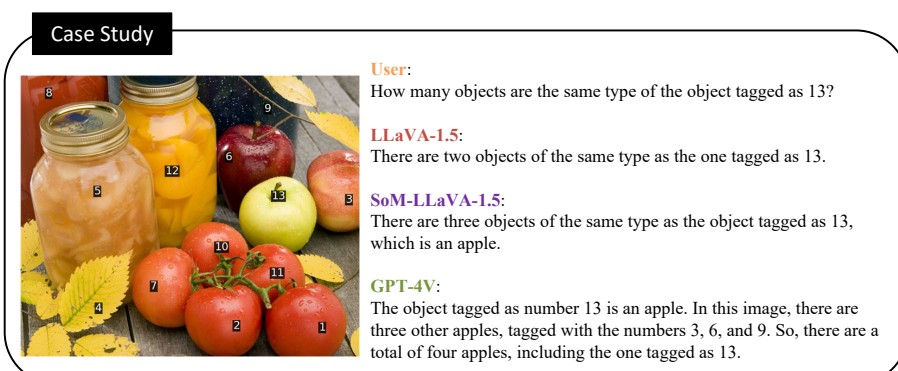

**Case Study**

**User**:
Whose pants' color is the same as someone else's white shirt?

**LLaVA-1.5**:
The man wearing brown pants has the same pants color as the woman wearing a white shirt.

**SoM-LLaVA-1.5**:
The person tagged with number 1 is wearing pants that are the same color as the person tagged with number 9's white shirt.

**GPT-4V**:
The person labeled with the number 2 is wearing white pants, and their color matches the white shirt of the person labeled with the number 9.

Figure 5: An example comparison for LLaVA, SoM-LLaVA and GPT-4V.

**Case Study**

**User**:
How many objects are the same type of the object tagged as 13?

**LLaVA-1.5**:
There are two objects of the same type as the one tagged as 13.

**SoM-LLaVA-1.5**:
There are three objects of the same type as the object tagged as 13, which is an apple.

**GPT-4V**:
The object tagged as number 13 is an apple. In this image, there are three other apples, tagged with the numbers 3, 6, and 9. So, there are a total of four apples, including the one tagged as 13.

Figure 6: An example comparison for LLaVA, SoM-LLaVA and GPT-4V.

## 6.2 Visual Reasoning with SoM Prompting

In this section, we present two examples of different models reasoning over the tagged images, aiming to show the potential of visual reasoning over tags, with a model that can understand SoM prompting.

In Figure 5, we examine a multi-step visual reasoning question (*i.e.*, "Whose pants' color is the same as someone else's white shirt"), which requires the MLLM to first identify the mentioned objects (*i.e.*, all the pants in the image and the white shirt), then compare their visual features (*i.e.*, the same white color) to answer the question. From Figure 5, we observe that LLaVA-1.5 provides an incorrect answer by falsely identifying the person wearing the white shirt as a female. This error can be caused by the inferior object recognition capacity in LLaVA-1.5, or the complexity of the multi-hop question. As for GPT-4V, it made a mistake by assigning tag-2 to the person on its right, leading to a wrong reasoning process and incorrect conclusion. In contrast, SoM-LLaVA-1.5 successfully identifies tags 1 and 9 with the same color in those image regions, while recognizing the two objects as white pants and white shirt, respectively. We show another example in Figure 6.

## 7 Conclusion

In this paper, we start with SoM prompting and propose a new learning paradigm for multimodal LLM training. We show that MLLMs can learn SoM prompting using a small set of synthetic data by listing items one by one. Moreover, we explore the broader impact and find our dataset can benefit general capabilities for MLLMs, where our enhanced model, SoM-LLaVA, consistently outperforms the original LLaVA model across seven multimodal benchmarks. Our dataset and models are released to facilitate vision and language research.

Overall, we hope this work could inspire future research on exploring new learning paradigms and data recipe for MLLM training, as well as ways to create better synthetic datasets in the era of LLMs.

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

# A Appendix

## A.1 Implementation details.

The LLaVA-1.5 model contains a CLIP-ViT-L-336px visual encoder (Radford et al., 2021) and a Vicuna-7/13B language model (Chiang et al., 2023), connected by an MLP projection layer. Our main experiments are conducted on 8X and 4X 80GB A100 GPUs for llava-13b and llava-7b models, with a batch size of 128 and 64, respectively. We train all models for 1 epoch, following hyperparameter setting in (Liu et al., 2023a).

We collected 10k SoM-listing data and 20k SoM-QA data using GPT-4V turbo. For visual tagging, we use the level-2 granularity of Semantic SAM to annotate all images from MS-COCO, to learn fine-grained object-text alignment. During inference, we find that the existing MLLM benchmarks mostly consist of high-level questions about an image, and level-1 annotation with fewer tags works better.

We report results of following MLLMs on public benchmarks: BLIP-2 (Li et al., 2023d), InstructBLIP (Dai et al., 2023), Fuyu-8B [1], LLaMA-Adapter-V2 (Gao et al., 2023), mPLUG-Owl-2 (Ye et al., 2023), Qwen-VL (Bai et al., 2023), SPHINX (Lin et al., 2023b), and LLaVA-1.5 (Liu et al., 2023a).

## A.2 Comparison Results on Reasoning on Images without Tags

We additionally analyze how LLaVA-1.5 and SoM-LLaVA-1.5 perform differently when images with no tags are provided. In Figure 7 and Figure 8 we can observe that the discrepancies between the attention maps extracted from the two models in both cases are relatively insignificant. Such observation suggests that LLaVA-1.5 has pre-trained with good multimodal cross-attention that enables the MLLM to capture the most characteristic visual features in the images. However, due to the lack of alignment between visual semantics and textual semantics, MLLMs like LLaVA-1.5 may not correctly associate textual information with relevant visual evidence, which further causes incorrect answers in visual reasoning. With SoM fine-tuning, we reinforce the MLLM's visual understanding of specific objects in the image by asking the model to list objects one by one. By bridging the objects' visual features and their semantic meanings, the MLLM can better refer to the visual objects and answer the questions with more accurate object descriptions.

## A.3 List Items One by One with SoM-LLaVa and GPT-4V

We present the open vocabulary listing results with our SoM-LLaVA and GPT-4V. As shown in Figure 9 and 10, our model is able to generate accurate descriptions of each tagged object, which learned the implicit tag-object association on images.

---

[1] https://www.adept.ai/blog/fuyu-8b

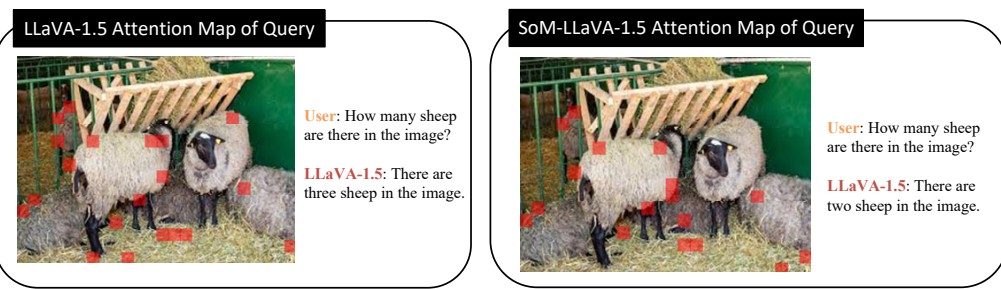

Figure 7: Attention map and visual question-answering comparative results from LLaVA-1.5 and SoM-LLaVA-1.5.

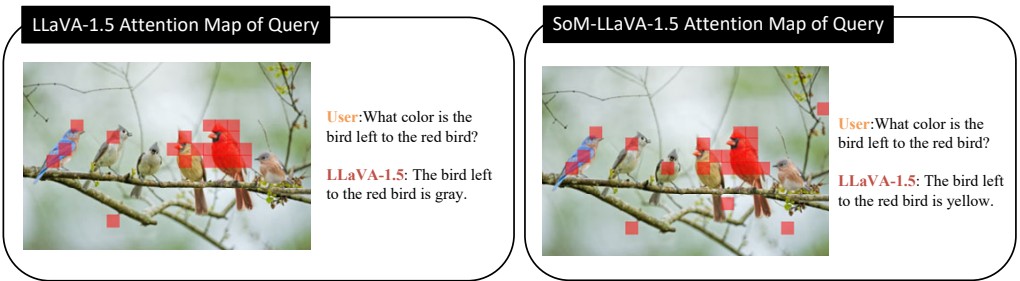

Figure 8: Attention map and visual question-answering comparative results from LLaVA-1.5 and SoM-LLaVA-1.5.

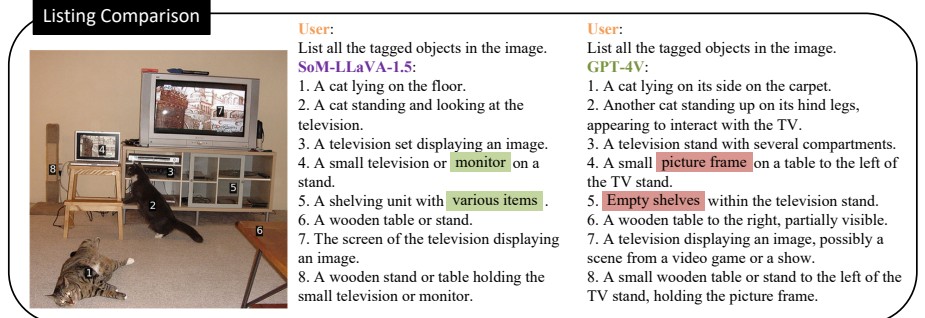

Figure 9: Open vocabulary listing with our model (SoM-LLaVA-1.5) and GPT-4V.

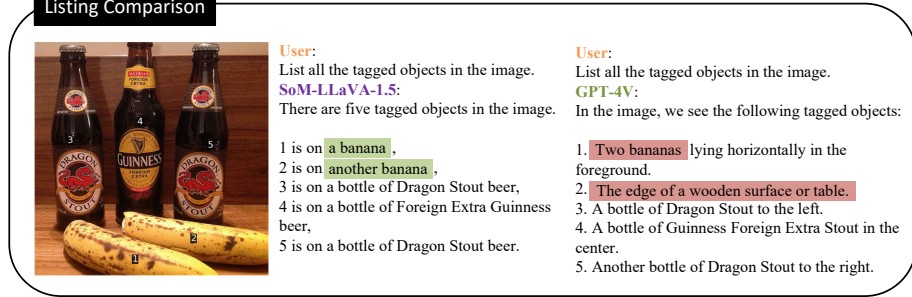

Figure 10: Open vocabulary listing with our model (SoM-LLaVA-1.5) and GPT-4V.

## A.4    GPT-4V listings with Different Prompting Methods

We present the listing results from GPT-4V with different prompting methods, as shown in Table 5 and Table 6. 2-shot in-context learning leads to more accurate listings.

| **Listing example from GPT-4V, woman by the water.** | |
| --- | --- |
| | 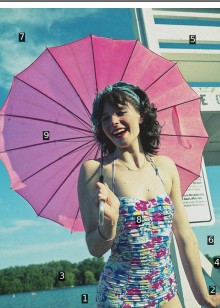 |
| GPT-4 zero-shot | - Tag 1 is on the water of a lake or river.  - Tag 2 is on a boat.  - Tag 3 is on a swimming platform or dock in the water. - Tag 4 is on the land or shore. - Tag 5 is on an informational sign or a notice board. - Tag 6 is on a tree. - Tag 7 is on a pink parasol or umbrella. - Tag 8 is on a bathing suit. - Tag 9 is placed where the blurred face is, describing no item. |
| GPT-4V improved sys-msg | Here are the items tagged in the image: 1. Blue water, possibly a lake or a river. 2. The lower part of a white, vertical pole, likely a part of the structure tagged with 5. 3. Green foliage or trees in the distance across the water. 4. The edge of a large white sign on a poll, with some text visible but not readable. 5. A white lifeguard stand or chair with blue accents and a sign attached. 6. Blue sky. 7. Blue sky with clouds. 8. The midsection of a person wearing a floral swimsuit. 9. A pink umbrella, held by the person whose face is blurred. The face of the person holding the umbrella is blurred for privacy reasons. |
| GPT-4V 2-shot learning | Here's a list of the tagged items in the image: 1. The bottom edge of a body of water, likely a lake or river given the surrounding context. 2. Lower arm and hand of a person holding an umbrella. The person's hand is specifically at the center of this tag. 3. Green foliage in the distance, on the other side of the water, which could be trees on the far shore. 4. The tag appears to be on the lower part of the torso of a person, specifically around the waist area. 5. A white structure with blue markings, which appears to be part of a lifeguard stand or beach-related signage. 6. A sign, partially obscured by the tag, with written text. 7. Clear blue sky. 8. The tag is centered on the midsection of a person wearing a floral swimsuit. 9. A pink parasol or umbrella, which the person is holding. |

Table 5: Examples of GPT-4V listings with zero-shot, improved system message, and 2-shot in-context learning.

## A.5 SoM Granularity Analysis

We present examples of visual tagging with semantic-SAM and different granularity, as shown in Figures 11 to 14.

## A.6 SoM Data in Existing Training Sources

Tables 7 to 9 shows a few examples that consist of listing in the text content.

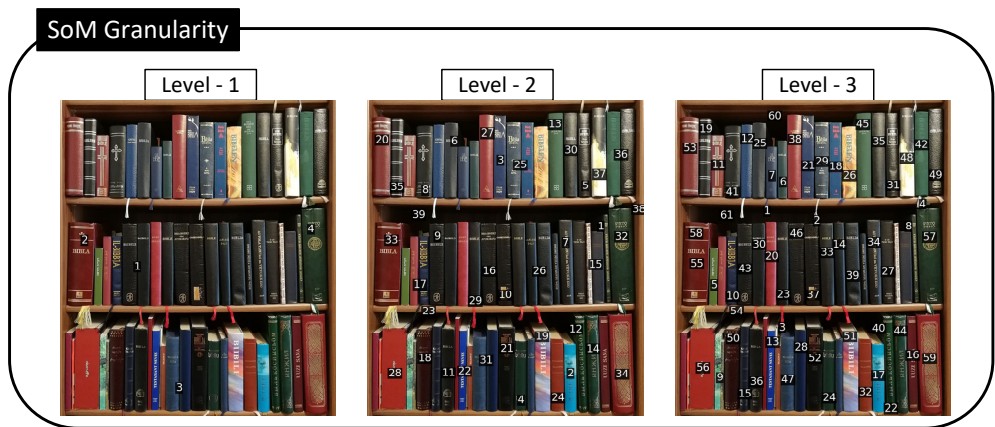

Figure 11: SoM tagging granularity analysis with level-1, level-2 and level-3 as coarse to fine.

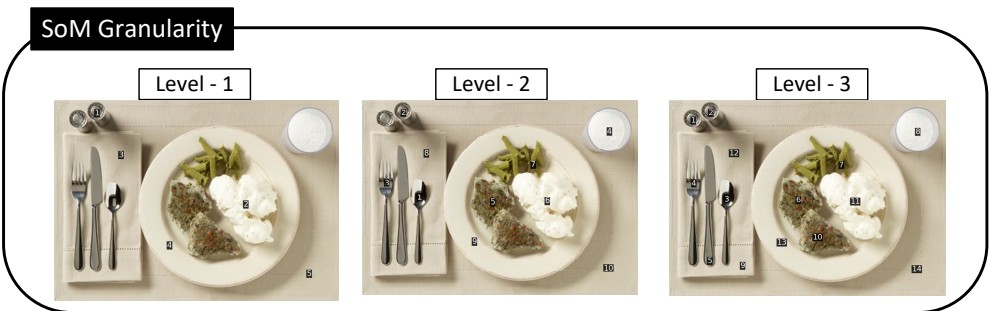

Figure 12: SoM tagging granularity analysis with level-1, level-2 and level-3 as coarse to fine.

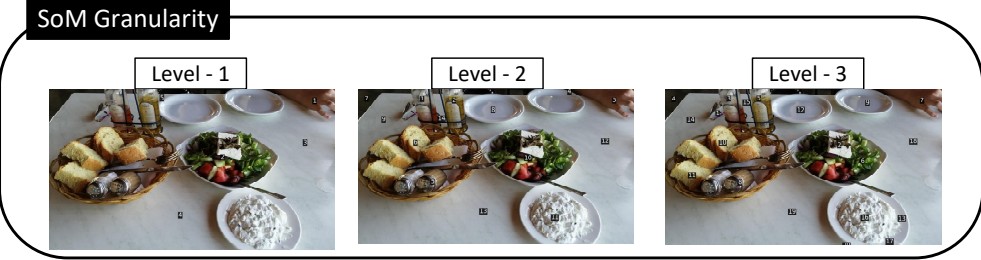

Figure 13: SoM tagging granularity analysis with level-1, level-2 and level-3 as coarse to fine.

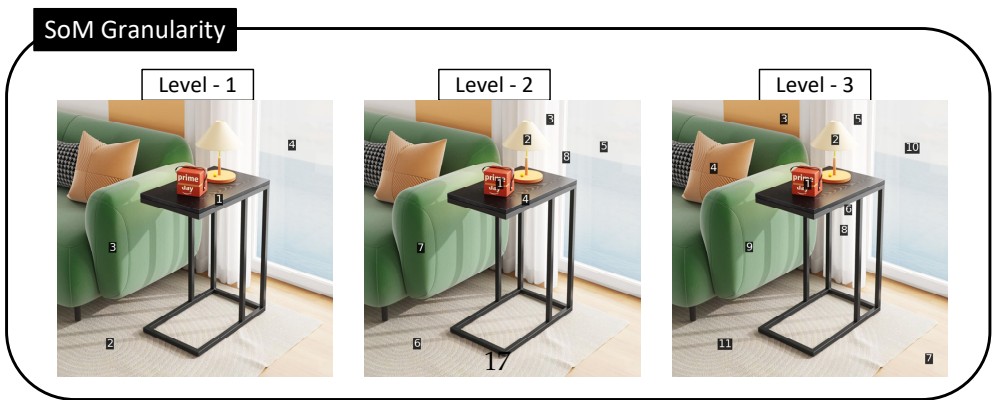

**Listing example from GPT-4V, flower and vase.**

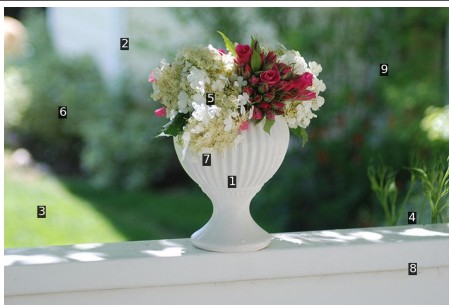

| | |
|---|---|
| GPT-4 zero-shot | 1. White vase 2. Small white flowers (species not clearly identifiable due to focus) 3. Green foliage or plant (specific species unclear) 4. Green plant (specific species unclear) 5. Plant stems (specific species unclear) 6. Green leaves (specific species unclear) 7. Green leaves (specific species unclear) 8. Green foliage or plant (specific species unclear) 9. Green foliage or plant in the background (specific species unclear due to focus) |
| GPT-4V improved sys-msg | 1. A white vase. 2. Part of a shrub or bush, possibly with tiny white flowers, in the background. 3. Greenery that looks like a lawn or ground cover, it's blurred and behind a ledge or step with tag number 8. 4. Some plants which are blurred, they could be grass or small plants. 5. Flowers, specifically white and possibly some type of hydrangea. 6. Green foliage of a tree or large bush in the background. 7. The pedestal or foot of the vase from tag number 1. 8. This tag seems to be positioned on the edge of a railing or a surface with a white finish. 9. More blurred greenery, possibly bushes or trees in the background. |
| GPT-4V 2-shot learning | 1. A white vase with a textured design. 2. Tag 2 is hovering above some sort of vegetation, likely a bush or shrub. 3. The tag is pointing to an area that appears to be part of a well-manicured lawn or grassy area. 4. Green grass or a plant is suggested by this tag, although it is not entirely clear. 5. This appears to be pointing to flowers in the bouquet, possibly white Hydrangeas. 6. Appears to be pointing to a shadow cast on the lawn or a shaded area of grass. 7. A cluster of flowers within the vase; they are likely white hydrangeas or another type of similar-looking flower. 8. Is tagging the flat surface which appears to be a ledge or balustrade, likely made out of painted wood based on the context. 9. A blurred area, suggestive of more green foliage or shrubbery in the background. |

Table 6: Examples of GPT-4V listings with zero-shot, improved system message, and 2-shot in-context learning.

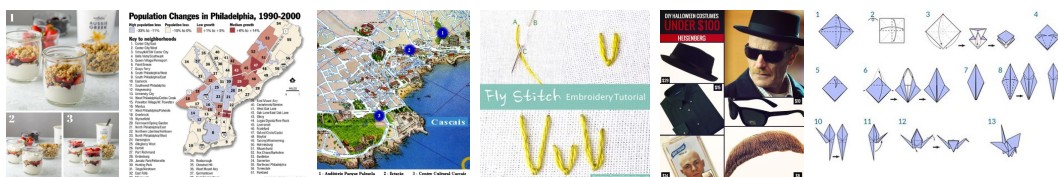

Figure 15: Discovered images with tagging annotations in LLaVA-Pretrain-LCS-558K.

| Image | Text |
|---|---|
| 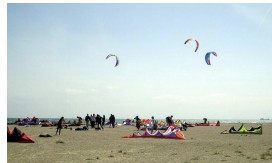 | **Question**: 
 What are some potential benefits or reasons people might gather at the beach to fly kites or engage in kiteboarding? |
| | **Answer**: 
 There are several reasons or benefits for people to gather at a beach to fly kites or engage in kiteboarding: 
 1. Socializing and bonding: Kite flying and kiteboarding events can bring people together, fostering a sense of community and social interaction among the participants. These activities allow people of all ages and backgrounds to establish connections and develop new friendships. 
 2. Outdoor recreation: Engaging in kite flying or kiteboarding can be a fun way to spend time outdoors and enjoy the fresh air and natural scenery. Beaches often offer wide-open spaces and consistent winds, which are ideal conditions for these activities. 
 3. Physical exercise: Both kite flying and kiteboarding can provide a form of physical exercise, improving the participants' overall fitness and well-being. Kite flying can be a low-intensity workout, strengthening the arms and shoulders, while kiteboarding is a more physically demanding sport that can enhance balance, core strength, and cardiovascular endurance. 
 4. Stress relief: Participating in kite flying or kiteboarding can help reduce stress and promote relaxation. These activities can serve as a form of mindfulness and mental exercise, allowing the participants to focus solely on the task at hand, temporarily setting aside everyday concerns and responsibilities. 
 5. Skill development: Flying kites and kiteboarding involve the development and refinement of various skills, such as coordination, dexterity, and control. Mastering these skills can provide a sense of accomplishment and personal growth. 
 In summary, people may gather at a beach to fly kites or engage in kiteboarding for various reasons, including socializing, outdoor recreation, exercise, stress relief, and skill development. These activities can offer a range of physical, mental, and social benefits, making them popular and enjoyable pastimes for those who take part. |

Table 7: An example from CogVLM-SFT-311K, with the answer text generated by GPT4-0314 and contains listing.

| Image | Text |
|---|---|
| 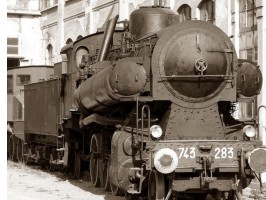 | **Question**: 
 Can you provide some background information about coal-powered locomotives? |
| | **Answer**: 
 Coal-powered locomotives, also known as steam locomotives, were the primary mode of railway transportation from the early 19th century until the mid-20th century. These locomotives used steam power generated by burning coal to operate their engines, which provided the force necessary for the train to move along the tracks. 
 A coal-powered locomotive was usually comprised of several key components: 
 1. A firebox or combustion chamber where coal was burned to create heat. 
 2. A boiler that contained water, heated by the firebox, to produce steam. 
 3. Pistons and cylinders that converted the steam pressure into mechanical motion, allowing the locomotive's wheels to turn. 
 4. A coal car or tender, which carried the coal fuel for the locomotive, usually located just behind the engine. 
 During the first half of the 20th century, coal-powered locomotives were steadily replaced by diesel and electric trains, which proved to be much more efficient and cleaner. Although steam locomotives are largely obsolete today, many are preserved in museums or operated for tourist attractions to provide a glimpse into the history of railway transportation. 
 The image of an old coal-powered locomotive and coal car on old tracks in a city setting offers a snapshot of a bygone era, showcasing the marvels of early railway technology and providing us with a window into the past. |

Table 8: An example from CogVLM-SFT-311K, with the answer text generated by GPT4-0314 and contains listing.

| Image | Text |
|---|---|
| 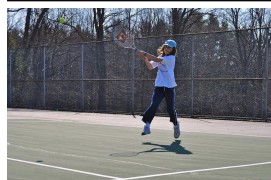 | **Question**:
Can you talk about the role of footwork, movement, and jumping in tennis? |
| | **Answer**:
Footwork, movement, and jumping are crucial components of a good tennis game, both in terms of enhancing overall performance and reducing the risk of injuries.
1. Footwork: Good footwork allows a tennis player to move efficiently and quickly around the court, enabling them to reach the ball and return it effectively. Proper footwork involves maintaining a balanced stance, taking small adjustment steps, and employing various movement patterns such as the shuffle, crossover, and side-step. Mastering footwork not only improves speed and agility but also ensures the player is in the right position to execute their strokes with optimal power and accuracy.
2. Movement: Movement is an essential aspect of tennis, as players must continually readjust their position to respond to their opponent's shots. Efficient movement on the court requires a combination of speed, agility, flexibility, and endurance. Anticipation plays a significant role in effective movement, as players must predict the trajectory of the ball and adjust their position accordingly. Top players often possess the ability to change direction quickly and cover the court effectively, enabling them to counter their opponent's strategies and make challenging shots.
3. Jumping: Jumping is a valuable skill in tennis, particularly when executing specific shots such as the overhead smash, serve or aggressive baseline shots. A well-executed jump allows the player to generate more power, reach for higher balls, and create sharper angles in their shots. Jumping while hitting the ball gives the player a slight height advantage, which enables them to strike the ball at a higher contact point, resulting in a more aggressive and effective shot. However, it is essential to practice proper techniques and ensure a safe landing to avoid potential injuries.
In conclusion, footwork, movement, and jumping are vital components of tennis performance. Mastering these skills makes a player more effective and efficient on the court, enhancing their ability to respond to various situations and execute a wide range of shots. Proper training and practice in these areas can significantly elevate a player's game and contribute to overall success in the sport. |

Table 9: An example from CogVLM-SFT-311K, with the answer text generated by GPT4-0314 and contains listing.

