# OpenReview forum: "List Items One by One: A New Data Source and Learning Paradigm for Multimodal LLMs"
_colmweb.org/COLM/2024/Conference — COLM_

### Official Review · Reviewer_gxtp · 2024-05-09

**Rating:** 7
**Confidence:** 4
**Ethics Flag:** 1

**Summary:**

The paper introduces a novel dataset and learning paradigm aimed at enhancing Multimodal Large Language Models (MLLMs) by teaching them to list items in images one by one. This method is designed to improve the association of visual objects with corresponding textual descriptions through a process called Set-of-Mark (SoM) Prompting. By tagging images with numeric identifiers and prompting models to list these tags, the study demonstrates significant improvements in visual reasoning and a reduction in hallucinations across several MLLM benchmarks.

**Questions To Authors:**

n/a

**Reasons To Accept:**

* Clear Motivation: The paper clearly articulates the motivation to enhance Multimodal Large Language Models (MLLMs) with the ability to list items, which could improve how these models handle visual objects in combination with textual descriptions.
* Innovative Approach: Introducing a novel dataset and a "list items one by one" learning paradigm is innovative. This method allows models to better integrate and process visual and textual information.
* Strong Results: The paper reports substantial benefits in terms of reduced hallucinations and improved visual reasoning across several MLLM benchmarks.

**Reasons To Reject:**

The research primarily focuses on t specific aspect of multimodal learning, which might be seen as a limited application area. Expanding this methodology to foundational models could make the work more impactful.

---

> ### Author Rebuttal · Authors · 2024-05-31
>
> Thank you for the appreciation of our work! Below are the response to your question.
>
> **Q1** The application area of this work and expanding this methodology to foundational models.
>
> **A1** Thanks for your suggestion. In our experiments, we have shown that our proposed dataset and learning paradigm is useful not only for empowering MLLMs with SoM prompting, but also for enhancing general visual reasoning and reducing hallucinations on popular benchmarks. It can be added as a complementary data source to existing visual instruction tuning datasets (see our additional results provided to other reviewers) for any visual LLM training.
>
> Nevertheless, it would certainly be interesting to explore how this methodology can apply to other types of foundational models, e.g., enhancing image/video generation for diffusion models.

---

### Official Review · Reviewer_v9mL · 2024-05-11

**Rating:** 5
**Confidence:** 4
**Ethics Flag:** 1

**Summary:**

This paper studies the Set-of-Mark capability of existing VLMs other than GPT4-V, and proposes a valid solution to construct training data for getting this capability.

**Questions To Authors:**

In section 3.2 it is unclear what is the binary classifier, and other details about the classifier.

**Reasons To Accept:**

1. Interesting problem of studying the Set-of-Mark capability of the VLM, the study from the paper indicates all VLMs can get similar capability if the model is trained specifically.

2. Promising but not the major contribution of the paper, the authors show that adding the training data designed for learning SoM capabilities can improve the VLM performance in general.

**Reasons To Reject:**

1. The contribution of the work is limited. It is yet another paper which constructs a training data, train the model, and get the capability.

2. The paper is not ready to publish yet. Many details about the approach are missing, I tried to read the section 4 multiple times but still cannot get the full picture of the construction process. More illustrations of the approach and the constructed examples will be necessary for reader to understand the proposed approach. Maybe move the examples from Appendix 1.6 into the main content. Instead, section 3 should go to Appendix, conjunction with moving the primary investigation results / conclusion goes to the introduction.


3. One key motivation is that the Set-of-Mark capability is important for visual referring prompting. As pointed in the related work, there is another important type of the referring technology which put user inputs (point, bounding boxes, sketch) directly into the text input. A comparison of Set-of-Mask approach and direct referring approach will make the paper more motivated. Also, the author should discuss and cite more papers here, there are many recent work on this direction.

---

> ### Author Rebuttal · Authors · 2024-05-31
>
> Thank you for reviewing our paper and the appreciation of it in terms of strengths. Below are our responses.
>
> **Q1** Contribution of the work.
>
> **A1** We would like to clarify our contributions here. First of all, data is perhaps one of the most critical aspects for today’s foundational models.
> Though we start with discussing SoM prompting and our work is motivated by it, we find “listing items one by one” is also effective for general MLLM training: if a model learns to list items with a specific order (in our case, it is determined by the visual numeric tags), it gains a fine-grained understanding of images.
> Verified by our experiments, we believe this is an important implication and opens up an interesting discussion on how to better align visual and language.
>
> **Q2** Details about the approach
>
> **A2** Thanks for the suggestion. Our data creation pipeline is simple, we add numeric tags to the images using semantic-SAM (section 4.1), and generate listing-style captions with GPT-4V (section 4.3). Data examples can be found in Figure.9-10 and Table.4 in appendix, we will add constructed examples to the main paper. We will also add a figure for readers to better understand the pipeline.
>
> **Q3** Comparison with direct referring
>
> **A3** In LLaVA-665k dataset we used, there is a category of visual referring data, ref-coco and visual genome, with 134k data. Hence our results in table.2 show that SoM data complements this type of data. Most visual referring LLMs [1,2] put coordinates of 1-2 objects into text, while we add visual tags on images to encode spatial information. We will cite more papers along this line, for example, [2, 3, 4]. Let us know if there are any specific papers to be included.
>
> **Q4** Details about the binary classifier
>
> **A4** We use the ViT from [5]. We use all raw images from coco train-2017, and construct tagged images by randomly adding 1-10 numeric tags (numbers chosen from 0-99) to the images. We built a balanced dataset with 236K images in total. We train the binary classifier for 5 epochs with learning rate 5e-5 and batch size 128. We will add details to the paper.
>
> [1] Improved Baselines with Visual Instruction Tuning
>
> [2] VisionLLM: Large Language Model is also an Open-Ended Decoder for Vision-Centric Tasks
>
> [3] Ferret: Refer and Ground Anything Anywhere at Any Granularity
>
> [4] Shikra: Unleashing Multimodal LLM’s Referential Dialogue Magic
>
> [5] An Image is Worth 16x16 Words: Transformers for Image Recognition at Scale

---

> > ### Author Response · Authors · 2024-06-06
> > **Author follow-up**
> >
> > Dear reviewer,
> >
> > Thank you again for reviewing our paper and providing useful feedback! Since the author-reviewer discussion is coming to an end, please let us know if there are any unresolved concerns. Moreover, we would be happy to hear your thoughts on our replies, and if you could consider increasing the score after reading other reviews and rebuttals.
> >
> > Thanks,
> > Authors

---

### Official Review · Reviewer_L6Ef · 2024-05-11

**Rating:** 7
**Confidence:** 5
**Ethics Flag:** 1

**Summary:**

This paper presents a new data source to better learn multimodal LLMs by collecting listing and QA data with set-of-mark (SoM) prompting from GPT4V. A new learning paradigm is also designed to enhance the visual understanding ability of MLLMs. Results show that the new dataset can improve the performance of MLLMs on various benchmarks and enable SoM prompting ability of MLLMs.

**Reasons To Accept:**

- Overall I think this is an interesting paper with some new insights and sufficient experimental study. The proposed method to collect SoM prompting data to improve MLLMs is well-motivated and interesting. Enabling the SoM prompting ability of MLLMs may be an effective method to close the gap between open-sourced MLLMs and GPT4V.

- Extensive experiments are conducted to show the effectiveness of the proposed method.

**Reasons To Reject:**

- The improvements look marginal on some of the benchmarks. In the abstract, it is mentioned that the method can " reduce hallucinations in Multimodal LLMs" but the improvements on POPE are not very significant. Can you provide an analysis of this phenomenon?

- How to ensure the quality of the dataset? GPT4V may produce wrong results. Would this affect the final performance of the models trained with the dataset?

---

> ### Author Rebuttal · Authors · 2024-05-31
>
> Thank you for the appreciation of our work! Below are the responses to your questions.
>
> **Q1** improvements on POPE are not very significant.
>
> **A1** The results are significant if we compare with the performance from other models (average F1 score), for example, the recently released phi-3-vision model has a score of 85.8, LLaVA-next-LLaMa-3 has a score of 87.0. Most models fall into the 83 to 87 range, even with more advanced models and data. The performance of LLaVA-1.5 is already decently high on POPE, hence it may be hard to improve further. Our SoM data can boost LLaVA-1.5 from 85.9 to 87.0, which shows its efficiency. We also present results from LLaVA-1.5-7B, where the performance improvements are consistently observed (slightly higher on some tasks, such as SEED-I and MM-Vet, than the 13B model)
>
> | Model       | POPE | MME  | SEED-I | LLaVA-W | MM-Vet |
> |-------------|------|------|--------|---------|--------|
> | LLaVA-1.5-7B          | 85.9    | 1510 |  64.8 | 63.4  | 30.5      |
> | SoM-LLaVA-1.5-7B | 86.5    | 1525 |  67.0 | 66.9  | 33.3      |
>
> **Q2** How to ensure the quality of the dataset.
>
> **A2** Thanks for the question. We explore two approaches, one is to modify system prompts, the other is to design high-quality demonstrations for few-shot prompting. We have an example in Table.4 in Appendix. We conduct qualitative comparison between our model and GPT-4V in Figure.9 and 10, showing our model can generate better listings than GPT-4V in a zero-shot setting. Overall, how to control the quality of any synthetic datasets is an interesting direction that requires further investigation.

---

> > ### Comment · Reviewer_L6Ef · 2024-06-06
> >
> > Thanks for your response. After reading other reviews as well as the rebuttal, I think the proposed SoM dataset and the results presented in the paper would be beneficial for future research. Therefore, I would upgrade my rating to "accept".

---

> > > ### Author Response · Authors · 2024-06-06
> > > **Author response**
> > >
> > > Thank you so much for the support of our work!

---

### Official Review · Reviewer_nfpB · 2024-05-14

**Rating:** 8
**Confidence:** 4
**Ethics Flag:** 1

**Summary:**

This paper presents a simple and compelling idea for enriching the spatial and fine-grained reasoning of multimodal large language models (MLLMs): rather than solely condition language generation on an image, the key idea is to augment an image with a “set of marks” (alphanumeric tags overlaid on the image) that are associated with key locations or reference points. As an example, given an image of a crowd at a concert, a plausible set of marks might include a different tag on each of the members of the crowd, a mark on the stage, and a mark on each of the musicians and their corresponding instruments.

Note that this idea of “marking” is an alternative input scheme that type checks with existing MLLMs that take in arbitrary images and generate text; at inference time, one can still provide untagged images, or they can manually tag an image with marks, and have the MLLM generate outputs referring to individual marked objects.

The bulk of the paper 1) discusses the failures of existing MLLMs such as LLaVa-1.5 when it comes to reasoning over “marked images”, 2) curating a small “marked” dataset of just 10-30K examples of images with labeled outputs just consisting of “List items one by one: <tag 1>: <description> ...”, and 3) evaluating MLLMs trained with this additional data on a spectrum of evaluation tasks.

In all, the proposed “marking” scheme is simple, scalable, and leads to MLLMs that outperform non-marked variants **in the absence of marking data during evaluation**, especially on tasks requiring fine-grained reasoning. The paper also presents case studies verifying that MLLMs trained with marking can integrate such marks for complex reasoning over full scenes, beyond the capabilities in the training data.

**Questions To Authors:**

Is there a reason that the results don’t include a 7B parameter model? Table 2 only shows a 13B variant of the mark-augmented models.

Do we need GPT-4V to produce the descriptions for each of the tagged objects? If we’re using MSCOCO images to seed our image dataset anyway, can we just use the object labels from that dataset (and possibly other datasets like Visual Genome that have individual region/object captions)?

**Reasons To Accept:**

This is a simple and scalable method for enriching arbitrary MLLMs to perform better at fine-grained reasoning tasks, and provides a new “input modality” for controllable interaction with MLLMs for more complex tasks. The work does a great job in identifying the limitations of existing MLLMs and MLLM datasets, and present a scalable approach for generating such “marked” data from existing image datasets (one that scales far beyond the data generated for this work).

The evaluations show that MLLMs trained with “marked” data not only can use such “marks” for complex reasoning, but also show that just training with this marked data enriches the abilities of MLLMs to perform fine-grained and spatial reasoning in the **absence of marks during evaluation** as well.

**Reasons To Reject:**

While the existing evaluations are compelling, it would make this work even stronger to explicitly evaluate the trained MLLMs on evaluations that provide calibrated/absolute metrics on spatial and compositional reasoning (e.g., GQA question answering accuracy, RefCOCO bounding box prediction, etc.). I recommend that the authors look into strengthening the existing evaluations for the final version of the paper.

---

> ### Author Rebuttal · Authors · 2024-05-31
>
> Thank you for the appreciation of our work! Below are the responses to your questions.
>
> **Q1** More evaluation benchmarks
>
> **A1** Thanks for the suggestion. We will test on more benchmarks before the final version.
>
> **Q2** Results on 7B parameter model
>
> **A2** Thanks for pointing it out. We are limited by compute resources when running ablations. Results from LLaVA-1.5-7B are here:
>
> | Model       | POPE | MME  | SEED-I | LLaVA-W | MM-Vet |
> |-------------|------|------|--------|---------|--------|
> | LLaVA-1.5-7B          | 85.9    | 1510 |  64.8 | 63.4  | 30.5      |
> | SoM-LLaVA-1.5-7B | 86.5    | 1525 |  67.0 | 66.9  | 33.3      |
>
> Additionally, we also present results from another recent M-LLM (xgen-mm-phi3 on HuggingFace, with a different architecture and pretrain recipe). We follow the setting on LLaVA to finetune the base model, with or without SoM data:
>
> | Model       | POPE | MME  | SEED-I | LLaVA-W | MM-Vet |
> |-------------|------|------|--------|---------|--------|
> | xgen-mm     | 86.4 | 1425 | 69.4   | 68.4    | 37.8   |
> | SoM-xgen-mm | 87.2 | 1437 | 71.2   | 72.0    | 39.9   |
>
> We will add these results to show the performance gain of different models with our SoM data
>
> **Q3** Do we need GPT-4V to produce the descriptions for each of the tagged objects?
>
> **A3** Thanks for initiating an interesting discussion. For MS-COCO we do have abundant annotations to create synthetic data. We considered this method in the early stage of our project, but after checking samples from VG, we find that the captions are not really “open-vocabulary” as what GPT-4V is capable of. For example, regional captions such as “man wearing in a hat”, “tree near the water” missing fine-grained attributes about the color of the hat, type of the tree, man’s posture or action, etc.
>
> A similar idea that may work for all image sources could be extracting local patches using object detectors, and use GPT-4V or other caption models to annotate the patches. In our work, we go with a simple and scalable method that relies on two foundation models, semantic-SAM and GPT-4V, to generate such data.
>
> For fair comparison with LLaVA-150k and share-GPT4v data, we use the same images from COCO to make sure no extra information are introduced in the images, so we can faithfully verify this new learning paradigm (list items one by one) is indeed effective. In the future, one can scale up the data to different image sources, using our pipeline or the alternatives discussed above.

---

> > ### Comment · Reviewer_nfpB · 2024-06-04
> > **Post-Rebuttal Response**
> >
> > Thank you for answering my questions and taking the time to add the additional 7B results; I thought this paper was very strong to begin with, and that opinion has not changed with the rebuttal - I will definitely be pushing for acceptance at this time!
> >
> > (Would definitely be great to run out the additional benchmarks for the final version though; just as a way to make the paper even stronger).

---

> > > ### Author Response · Authors · 2024-06-04
> > > **Author Response**
> > >
> > > Thank you so much for the appreciation and support of our work!
> > >
> > > Here are the results on GQA ( it takes longer than expected to add GQA results with some bug fixes). We will continue exploring additional benchmarks to show the effectiveness of our method.
> > >
> > > | Model             | GQA  |
> > > |-------------------|------|
> > > | LLaVA-1.5-7B      | 62.0 |
> > > | SoM-LLaVA-1.5-7B  | 62.7 |
> > > | LLaVA-1.5-13B     | 63.3 |
> > > | SoM-LLaVA-1.5-13B | 63.8 |

---

### Official Review · Reviewer_YAe1 · 2024-05-14

**Rating:** 6
**Confidence:** 4
**Ethics Flag:** 1

**Summary:**

The paper proposes a new dataset for Set-of-Mark (SoM) prompting for MLLMs. SoM allows MLLMs to associate objects with text by assigning numbers to different objects. Adding the proposed dataset to existing visual instruction tuning datasets improves the performance on five visual reasoning tasks. The SoM dataset is created by modifying an existing dataset (MS-COCO) that has human annotations corresponding to bounding boxes, and captions. An existing model (SAM) is used to segment the images prior to adding numbers on top of each segment then use GPT4-v to generate the textual descriptions. The authors fine-tune the pertained LLaVA-1.5 model using LLAVA-1.5 fine-tuning data in addition to the newly created dataset and show improvements on five visual reasoning tasks.

**Reasons To Accept:**

- Interesting approach & well though-out analysis on using SoM data to improve the  visual reasoning performance of MLLMs.
- Fine-tuning an MLLM using the proposed dataset improves the performance on five visual reasoning tasks.

**Reasons To Reject:**

- The proposed approach and dataset are only applied to one MLLM. It would be good to try other MLLMs to see if the results generalize.
- It would be good to evaluate the in-context learning (without fine-tuning) performance of using exemplars from the new dataset.

---

> ### Author Rebuttal · Authors · 2024-05-31
>
> Thank you for the appreciation of our work! Below are the responses to your questions.
>
> **Q1** The proposed approach and dataset are only applied to one MLLM. It would be good to try other MLLMs to see if the results generalize.
>
> **A1** Thanks for your suggestion. We deliver additional results from two other MLLMs.
>
> First, we used another recently open-sourced M-LLM with pretrained weights available (xgen-mm-phi3-mini-base-r-v1 on HuggingFace, with a different model architecture and pretrain data compared with LLaVA). We compare with two visual instruction tuning recipes as in our paper: LLaVA-665K and SoM-LLaVA-695k to finetune the same base model, and evaluate their performance on the five benchmarks, using raw images without tags during testing. The performance comparison is shown below:
>
> | Model       | POPE | MME  | SEED-I | LLaVA-W | MM-Vet |
> |-------------|------|------|--------|---------|--------|
> | xgen-mm     | 86.4 | 1425 | 69.4   | 68.4    | 37.8   |
> | SoM-xgen-mm | 87.2 | 1437 | 71.2   | 72.0    | 39.9   |
>
> We also tested our dataset with LLaVA-1.5-7B which has a smaller language model (vicuna-7B) compared to the one we reported in Table.2. The performance comparison is shown below:
>
> | Model       | POPE | MME  | SEED-I | LLaVA-W | MM-Vet |
> |-------------|------|------|--------|---------|--------|
> | LLaVA-1.5-7B          | 85.9    | 1510 |  64.8 | 63.4  | 30.5      |
> | SoM-LLaVA-1.5-7B | 86.5    | 1525 |  67.0 | 66.9  | 33.3      |
>
>
> **Q2** It would be good to evaluate the in-context learning (without fine-tuning) performance using exemplars from the new dataset.
>
> **A2** Thanks, that is an interesting direction to explore, with the recent releases of open-sourced interleaved M-LLMs that are capable of performing in-context learning.

---

> > ### Author Response · Authors · 2024-06-06
> > **Author follow-up**
> >
> > Dear reviewer,
> >
> > Thank you again for reviewing our paper and providing useful feedback! Since the author-reviewer discussion is coming to an end, please let us know if there are any unresolved concerns. Moreover, we would be happy to hear your thoughts on our replies, especially the additional results on llava-1.5-7b and xgen-mm, and if you could consider increasing the score given our responses above.
> >
> > Thanks,
> > Authors

---

### Official Review · Reviewer_HXE7 · 2024-05-23

**Rating:** 8
**Confidence:** 3
**Ethics Flag:** 1

**Summary:**

Set-of-Mark (SoM) prompting (placing numbered tags on images before feeding them to the models) has been shown to improve visual reasoning on GPT-4V.
This paper creates a dataset helping MLLMs to acquire the SoM visual prompting ability, and shows that MLLMs other than GPT-4V (such as LLaVA-1.5) can also benefit from this technique. The SoM models are evaluated on 5 benchmarks and show improvement even without the tags. The authors also investigated the mechanism of SoM.

**Questions To Authors:**

In Figure 3 (a), increasing the listing data from 75k to 100k seems to lead to a slightly lower accuracy for both the 7B and 13B models. Do the authors have an idea of why?

Adding tags may obstruct very small objects. Can the technique still be helpful in this scenario?

Can the authors indicate how long the training was for both models?

The LaTeX style seems different from the official version.

Typo:
- Paragraph 1: "can implicitly aligning [...]" -> "can implicitly align [...]"
- 4.3, "GPT-4V is asked to generate mulit-turn" -> "multi-turn"

Minor:
- Figure 2: it's a bit hard to read the tags behind the attention maps, they can be emphasized. Also, some red patches seem to be going a bit off the image.

**Reasons To Accept:**

Interesting work, with three-fold contributions: probing SoM's mechanism, proposing an approach adding the SoM capability for other MLLMs with a dedicated dataset. The work's insight can be beneficial for the research community.

The paper is clearly written, well structured, and easy to follow. The appendix also contains useful details.

The authors stated that the dataset and models will be released.

**Reasons To Reject:**

The approach's effectiveness is only demonstrated with LLaVA-1.5. Probing other types or sizes of VLMs can be useful to better understand its applicability.

This adds additional overhead and needs to retrain the instruction tuning stage. It seems unclear if fine-tuning an already instruction-tuned model using the SoM data would work.

---

> ### Author Rebuttal · Authors · 2024-05-30
>
> Thank you for the appreciation of our work! Below are the responses to your questions.
>
> **Q1** Effectiveness on other types or sizes of VLMs
>
> **A1** Thanks for your suggestion. The reason we choose LLaVA is, it is a fully open-sourced model (both data and model weights for both pretraining and SFT stages are released), so it makes the full diagnosis possible. And its model size and training recipe is what we can afford given the limited budget.
>
> Here we deliver additional results from another recently open-sourced M-LLM (xgen-mm-phi3-mini-base on HuggingFace, with a different architecture and pretrain recipe). We use two datasets: LLaVA-665K and SoM-LLaVA-695k to finetune the same base model, and evaluate their performance:
>
> The performance comparison is shown below:
> | Model       | POPE | MME  | SEED-I | LLaVA-W | MM-Vet |
> |-------------|------|------|--------|---------|--------|
> | xgen-mm     | 86.4 | 1425 | 69.4   | 68.4    | 37.8   |
> | SoM-xgen-mm | 87.2 | 1437 | 71.2   | 72.0    | 39.9   |
>
> We also tested our dataset with LLaVA-1.5-7B with a smaller LM (vicuna-7B), as shown below:
> | Model       | POPE | MME  | SEED-I | LLaVA-W | MM-Vet |
> |-------------|------|------|--------|---------|--------|
> | LLaVA-1.5-7B          | 85.9    | 1510 |  64.8 | 63.4  | 30.5      |
> | SoM-LLaVA-1.5-7B | 86.5    | 1525 |  67.0 | 66.9  | 33.3      |
>
> **Q2** Additional overhead for visual instruction tuning.
>
> **A2** Our dataset is relatively small (30K in total), adding less than 5% training time for finetuning. Finetuning an already instruction-tuned model with SoM data is possible, though it may lead to forgetting of previous data. We follow the common practice to mix the training data.
>
> **Q3** listing data from 75k to 100k
>
> **A3** We generate the listing with coco annotations. The object classes are limited (80 classes) hence there could be some overfitting issue. Also, the annotation itself can be noisy, so a matching score of around 85% may be hard to improve. We will clarify this in the revision.
>
> **Q4** Adding tags to small objects
>
> **A4** We follow the tagging method in the SoM paper [1] where we slightly move the tag down from the center by a few pixels, so small objects are still visible.
>
> **Q5** Training time
>
> **A5** It takes around 20-22 hours to train LLaVA-1.5-13B on 8 A100s (80GB).
>
> **Q6** Typo and formatting.
>
> **A6** Thanks for your careful review, we will revise the typos.
>
> [1] Set-of-Mark Prompting Unleashes Extraordinary Visual Grounding in GPT-4V

---

> > ### Comment · Reviewer_HXE7 · 2024-06-05
> > **Reponse to the authors**
> >
> > Thanks to the authors for the details and additional results on other open-sourced base models, showing consistent improvement. I believe that adding the details and results provided can make the paper clearer.
> >
> > I increased my score as the answers dealt with my main concerns.

---

> > > ### Author Response · Authors · 2024-06-05
> > > **Author response**
> > >
> > > Thank you so much for the appreciation and support of our work, and the prompt response!

---

### Decision · Program_Chairs · 2024-07-10

**Decision:**

Accept

**Comment:**

This paper examines whether multimodal large language models (MLLM) can map objects "marked" in the image to the words in a prompt. The authors create a dataset based on MSCOCO by automatically finding the objects, tagging them with numbers ("marking them"), and adding captions using a LLM. They also discuss how a recent model fail at describing the marked aspects of an image, and show that the model performance can be improved when trained on the collected dataset. The reviewers find the paper clear, and the proposed approach interesting and well executed. I agree with the reviewers but encourage the authors to evaluate their method on commonly-used fine-grained benchmarks such as GQA (as Reviewer nfpB points out) as well as typical retrieval and QA benchmark to see if the performance on generic tasks changes/degrades with the additional training.

[At least one review was discounted during the decision process due to quality]